# Current Research Provides Insight into the Biological Basis and Diagnostic Potential for Myalgic Encephalomyelitis/Chronic Fatigue Syndrome (ME/CFS)

**DOI:** 10.3390/diagnostics9030073

**Published:** 2019-07-10

**Authors:** Eiren Sweetman, Alex Noble, Christina Edgar, Angus Mackay, Amber Helliwell, Rosamund Vallings, Margaret Ryan, Warren Tate

**Affiliations:** 1Department of Biochemistry, University of Otago, Dunedin 9016, New Zealand; 2Howick Health and Medical Centre, Auckland 2014, New Zealand; 3Department of Anatomy, University of Otago, Dunedin 9016, New Zealand

**Keywords:** myalgic encephalomyelitis, chronic fatigue syndrome, diagnostic biomarker, inflammation and immunity, metabolism, mitochondria, circadian rhythm, neuro-inflammation

## Abstract

Myalgic encephalomyelitis/chronic fatigue syndrome (ME/CFS) is a severe fatigue illness that occurs most commonly following a viral infection, but other physiological triggers are also implicated. It has a profound long-term impact on the life of the affected person. ME/CFS is diagnosed primarily by the exclusion of other fatigue illnesses, but the availability of multiple case definitions for ME/CFS has complicated diagnosis for clinicians. There has been ongoing controversy over the nature of ME/CFS, but a recent detailed report from the Institute of Medicine (Academy of Sciences, USA) concluded that ME/CFS is a medical, not psychiatric illness. Importantly, aspects of the biological basis of the ongoing disease have been revealed over the last 2–3 years that promise new leads towards an effective clinical diagnostic test that may have a general application. Our detailed molecular studies with a preclinical study of ME/CFS patients, along with the complementary research of others, have reported an elevation of inflammatory and immune processes, ongoing neuro-inflammation, and decreases in general metabolism and mitochondrial function for energy production in ME/CFS, which contribute to the ongoing remitting/relapsing etiology of the illness. These biological changes have generated potential molecular biomarkers for use in diagnostic ME/CFS testing.

## 1. Introduction

### Myalgic Encephalomyelitis/Chronic Fatigue Syndrome (ME/CFS): A Significant Global Health Problem

The reported worldwide prevalence of ME/CFS varies from 0.4–2.6% of the population among countries and cultures [1], making it significantly more prevalent than other fatigue illnesses, such as multiple sclerosis, with, for example, up to ~240,000 Australians [2] and ~20,000 New Zealanders [3] affected with ME/CFS compared to ~25,000 and ~4000 respectively with multiple sclerosis. The first recognition of ME/CFS in New Zealand dates from an outbreak in a small rural town, Tapanui, in 1983, with an unexplained flu-like illness given the name ‘Tapanui flu’ [4], which was later classified as ME/CFS [5]. Similar outbreaks have been reported in isolated communities around the world since the 1930s [6]. A 2013 meta-analysis [7] found that ME/CFS prevalence from self-reporting assessment was 3.28% (95% Confidence Interval (CI): 2.24–4.33) but by clinical assessment was only 0.76% (95% CI: 0.23–1.29), using the 1994 Fukuda clinical case definition [8]. This discrepancy in prevalence highlights the importance of an understanding of the disease by health practitioners, and the use of clinically consistent diagnostic criteria for ME/CFS. 

## 2. Clinical Characteristics of ME/CFS

Since ME/CFS has as yet no conclusive diagnostic laboratory test, and an ill-defined pathophysiology [9], there has been a diverse range of opinions as to the precise nature of the disease among health professionals and throughout wider society. This confusion surrounding the ME/CFS diagnosis among health practitioners worldwide has meant that patients and families are often without the support of their healthcare system and social support systems. A diagnostic molecular biomarker, tool, or accessible procedure specific for ME/CFS, that is readily transferable to diagnostic laboratories for routine tests on community-wide patient samples, is urgently needed.

Onset of ME/CFS frequently follows an acute viral infection or period of stress [1], but more gradual onset can occur, with the complex of symptoms developing over a period of weeks or months [10]. Many unique ‘outbreaks’ of an ME/CFS-like disease are recorded, suggesting it can arise from the spread of an initiating infectious agent. Certainly, ME/CFS is commonly self-reported following a glandular fever episode from Epstein Barr virus infection [11,12,13,14]. Twin studies indicate a genetic susceptibility for ME/CFS, with a higher rate of fatigue concordance in monozygotic twins than dizygotic twins [15,16,17]. In susceptible people, a diverse range of initiating agents can produce the same physiological ‘shock’ that precipitates progression into the life-long condition of ME/CFS [18]. Apart from viruses, factors such as toxins or agricultural chemicals like organophosphates, and physiological stressors, such as vaccinations [11,18], can precipitate the illness [11,12,13]. A recent publication examined 19 cases of patients diagnosed with either ME/CFS or fibromyalgia following hepatitis B vaccination, concluding that, in some cases, both of the illnesses could be temporally related to immunisation as part of autoimmune (auto-inflammatory) syndromes induced by adjuvants (ASIA) [19]. ME/CFS affects people of all ages and within all socio-economic groups, but it is more common in women (reported to be a ratio between 2:1 to 6:1, female to male) [11,12,13]. The defining symptom of ME/CFS is persistent, debilitating fatigue, lasting beyond six months. Most clinical diagnostic criteria describe this as physical and mental and disabling, usually of acute onset. It is significantly exacerbated by exercise, and mental or emotional exertion (post-exertional malaise), and is not alleviated by rest [6,14,20]. A myriad of flu-like and respiratory symptoms, cognitive impairment, tender lymph nodes, muscle and joint pain (myalgia), severe headaches, new allergies, severely disturbed sleep patterns with un-refreshing sleep, and mood changes are commonly experienced [6,14,20]. Multi-system co-morbidities, for example, POTS (postural orthostatic tachycardia syndrome), depression, and irritable bowel syndrome are often found [20,21]. Nevertheless, the severity and the range of ME/CFS symptoms can vary, with three in every four patients progressing from an extended acute phase to a chronic state of ongoing debilitating illness that still requires dramatic lifestyle changes to manage the frequency of severe relapses [14]. It is claimed that only about 5% of ME/CFS patients will return to their previous state of health and well-being [22], therefore for most of those affected it is a life-long disease.

## 3. Clinical Case Definitions

At present, a formal diagnosis is given only after eliminating all other diseases with similar symptoms, and with the presence of a range of self-reported symptoms fitting within defined sets of clinical criteria [9,11,12,20,21]. The difficulty for both patients and health practitioners has been that over 20 different case definitions or diagnostic criteria for ME/CFS exist [23]. Since the underlying pathophysiology of ME/CFS is still largely unknown, there is no gold standard against which to assess the effectiveness of each case definition. The 1994 Fukuda diagnostic criteria [8] developed by the Centre for Disease Control in the USA, is most commonly used by researchers and clinicians [1], yet it does not include the core defining symptoms of post-exertional malaise and neurocognitive disturbances, nor does it exclude patients whose symptoms may originate from a psychiatric disorder. The Canadian Consensus Criteria (CCC) [24] developed in 2003 by an international ME/CFS expert group was a significant improvement as it highlighted post-exertional malaise as a core symptom, along with fatigue, sleep dysfunction and pain. Additionally, neurological/cognitive and autonomic/neuroendocrine/immune symptom groups were included. In 2011, the ‘International Consensus Criteria’ were formulated as a refinement of the CCC, putting emphasis on inflammation and neuropathology and focusing on neurological disturbance, immune/gastrointestinal, and energy impairments [20]. These criteria have yet to be generally accepted, and they may be selective for a subset of ME/CFS patients only. 

To redress the confusion created by so many case definitions, a clinical guideline IACFS/ME primer for General Health Practitioners (GPs) was developed in 2012 by a panel of experts from the International Association for Chronic Fatigue Syndrome/Myalgic Encephalomyelitis (IACFS/ME) [21]. This included commonly used clinical guidelines, but showed considerable variation in symptoms and co-morbidities from those based on individual case definitions [25]. A detailed review of the criteria used to diagnose ME/CFS was released in 2015 by the Institute of Medicine (IOM) of the Academy of Sciences USA [6] along with a simplified core set of diagnostic criteria. The IOM report acknowledged that the stigma associated with a diagnosis of ME/CFS is largely due to both the lack of knowledge of the disease and the lack of acknowledgement of it as a distinct disease. Most importantly, the report stressed that based on all the available evidence, ME/CFS is a medical and not a psychiatric illness [6]. In our view, the Canadian Consensus Criteria (CCC) [24] are the best available definitions for both clinical diagnosis and for preclinical patient research studies. 

## 4. Physiological Cause of ME/CFS and Current Treatments

To date, treatments for ME/CFS have targeted specific symptoms, such as sleep disruption, fatigue, muscle pain and emotional disturbance [11]. While the underlying primary physiological deficit in ME/CFS is still unknown, many differences in physiology and metabolism between ME/CFS patients and healthy controls have now been discovered in the last 2–3 years, enabling some understanding of the biological basis for the severely compromised health of ME/CFS patients. The research has highlighted areas for potential treatment. As ME/CFS patients have a significant immunological dysfunction, intravenous immunoglobulin therapy, interferon and ampligen, have been explored, but as yet with no conclusive outcomes [1]. Antidepressants, anti-allergy drugs, vitamin and mineral supplementation, and non-steroidal anti-inflammatory drugs have all been trialed among patient groups with mixed results [11,26]. A chance finding of remission of ME/CFS symptoms in a patient within a group undergoing cytotoxic treatment for Hodgkin’s Lymphoma resulted in the hypothesis that B-cell depletion might provide a potentially effective treatment for ME/CFS [27]. Subsequent trials with ME/CFS patients using a monoclonal anti-CD20 antibody targeting a B-cell surface protein, Rituximab, seemed highly promising [28,29]. Disappointingly, however, the multicentre phase III trial ultimately was less successful [30]. Most recently, a phase II trial of a mixture of the Central Nervous System (CNS) stimulant Ritalin (methylphenidate hydrochloride) and mitochondrial support nutrients (KPAX002) [31] suggested that there was a trend towards improvement in fatigue. Two behavioural interventions, graded exercise therapy (GET) and cognitive behavioural therapy (CBT), have been controversial treatments for ME/CFS [11]. GET focuses on gradually increasing physical activity over time, but it has had limited success and often exacerbates the characteristic ‘exercise intolerance’ or post-exertional malaise of ME/CFS, as well as other symptoms. CBT by contrast is a psychotherapy approach that encourages patients to analyse their symptoms and develop strategies to function around them. Undoubtedly this approach has benefitted some patients in managing and living with their disease. A large-scale ‘**P**acing, graded **A**ctivity, and **C**ognitive behaviour therapy; a randomised **E**valuation’ (PACE) study incorporating these strategies in the UK was claimed to show a favourable response with both GET and CBT interventions together [32], but the method of analysis and thereby the conclusions have been strongly criticised [33], particularly in a special issue in 2017 of the Journal of Health Psychology [34].

## 5. Biomarkers Leading to a Diagnostic Test

As there is no single molecular biomarker test for ME/CFS, there are long delays and high costs involved in the diagnostic process, with increased potential for misdiagnosis, all of which fundamentally impedes patient care. Many potential diagnostic biomarkers have been identified by researchers—almost all of which indicate the involvement of improper immune function, inflammation, and signs of autoimmunity, e.g., differences in cytokine profiles, natural killer (NK) cell function, or responsiveness of T-cells, in ME/CFS. To date, research into clinically useful diagnostic biomarker identification for ME/CFS has been limited generally to small cohorts (with study sizes frequently <10, and rarely with validation in larger cohorts above *n* = 40). A further limitation to biomarker discovery is the lack of any comprehensive follow-up studies of potential biomarkers with different ME/CFS patient groups, or comparing ME/CFS with other similarly presenting illnesses. Another important factor obstructing biomarker discovery is the use of different diagnostic criteria from the many available to define the ME/CFS patient group, preventing meaningful comparisons between studies. While the majority of research groups do use the 1994 Fukuda criteria, as discussed earlier, the Fukuda criteria may confound clinical or diagnostic biomarker studies as it imprecisely defines ME/CFS symptomology and fails to exclude patients with a psychiatric disorder. The lack of follow-up studies, or failure to validate the results of an original study, has meant that potential biomarkers have rarely progressed to clinical trials. 

Despite these significant handicaps to diagnostic research, exciting recent studies have emerged that seem to have considerable promise for further development into a general accessible diagnostic test. Nanoneedle bioarray technology, developed by Professor Ron Davis and colleagues at Stanford, measures a unique impedance signature that can differentiate moderate to severe ME/CFS sufferers (*n* = 20) from healthy controls (*n* = 20) [35]. The nanoneedle measures electrical impedance modulations resulting from cellular or molecular interactions in response to an induced high salt stressor. The test was able to differentiate ME/CFS from healthy controls from peripheral blood mononuclear cells (PBMCs) and plasma samples. The origin of the distinct different impedance signature in the ME/CFS group has not yet been identified, but the authors suggest that this may be caused by the Na/K ATPase transmembrane ion pump in ME/CFS cells, or a potential size change in cells as a result of increased osmotic pressure, or a change in the composition of cell membranes in response to the stressor. As yet, this technology has not been shown to distinguish ME/CFS from other related illnesses, and this will take time to resolve. 

## 6. Changes in the Biology of ME/CFS Patients

Emerging molecular technologies have enabled significant insights into the metabolic and physiological abnormalities that sustain ME/CFS. We have applied these technologies in an ongoing preclinical analysis with a group of 10 ME/CFS patients, diagnosed using the CCC criteria, and matched controls for study according to the principles of precision medicine [36] to obtain an integrated ‘molecular picture’ of the illness. We have collected cytokine (Bio-Plex Human Cytokine 27-plex Assay) [37] and microRNA [38] expression data from patient and control plasma (TaqMan miRNA array), and also genes (RNAseq transcriptome [~13,000 gene transcripts] [38], and SWATH-MSprotein expression data [~1800 proteins], publication in preparation) from peripheral blood mononuclear cells (PBMCs) [38]. Statistically significant changes were identified, despite the small size of the study group with age/gender matched controls. Despite patient heterogeneity in age, gender, and stage of illness, similar patterns of changes in specific processes and pathways were observed. We identified significant dysregulation of immune/inflammatory pathways and oxidative stress linked to metabolic and mitochondrial dysfunction. Immune, inflammatory, cytokine and apoptosis pathways were enhanced, while mitochondrial function, general cellular metabolic and lipid metabolic pathways were suppressed [38]. These findings are consistent with the emerging data from other ME/CFS studies, some with larger cohorts (Table 1) [38,39,40,41,42,43,44,45,46,47,48,49], and show the utility of the approach utilizing precision medicine to elucidate disease pathology in small patient numbers—a practice used successfully in studies of rare diseases [50].

In particular, our transcriptome study [49] identified the top three upregulated genes in the ME/CFS group, as*IL8*, *NFKBIA* and *TNFAIP3* (see Figure 1), all of which are early-responders to Tumour Necrosis Factor (TNF)-induced **N**uclear **F**actor **k**appa-light-chain-enhancer of activated **B** cells (NF-κB) activation [51]. 

Increased *IL8* expression occurs as a result of TNF-induced NF-κB activation, and the proteins A20 (*TNFAIP3*) and **N**uclear **F**actor of Kappa light chain polypeptide gene enhancer in **B** cells **I**nhibitor, **A**lpha (*NFKBIA*) are part of the two main negative feedback loops of NF-κB-driven transcription [51]. Furthermore, TNFα is a potent inducer of IL-8 secretion, through a transcriptional mechanism regulated by NF-κB. Indeed, increases in IL-8 and TNFα have been identified in several ME/CFS cytokine and immune studies. Chronic inflammation is also amplified through the NF-κB signaling pathway. The increase in expression of these three gene transcripts in the ME/CFS group implies that there is an ongoing biological inflammatory response, and a counter-response to the unwanted excess activity of NF-κB and inflammation in ME/CFS, driven by TNFα. 

With the same study group, we have investigated the abnormal activation of protein kinase RNA-activated (PKR) as a potential biomarker for ME/CFS. This kinase has been described as a ‘universal immunological abnormality’ in ME/CFS [52]. ME/CFS often follows an acute viral infection, suggesting that the key role PKR plays in the innate immune response to infection may be significant in ME/CFS symptomology. The efficacy of PKR as a diagnostic biomarker for ME/CFS results from the fact that PKR is phosphorylated when activated. Healthy controls had undetectable phosphorylated PKR in protein extracts of PBMC cells using an in-house affinity purified antibody (two stage purification-positive and negative affinity steps). Phosphorylated PKR (pPKR) was in contrast detected in the protein cell extracts of ME/CFS patients. A ratio of pPKR to inactive unphosphorylated PKR examined between the patients and controls revealed differences between the two groups (see Figure 2). 

It should be noted that not all patients scored positive for pPKR, but the study group on average had suffered with their illness for 12 years. It would be important to re-evaluate the pPKR/PKR ratio in larger patient groups undergoing diagnosis at the *early* stage of their illness to determine false negative rates. Then, it will be clear whether PKR has significant promise as a diagnostic tool.

## 7. Global Research into ME/CFS Biology

### 7.1. Recent Research Studies Have Focused on Several Key Areas

#### 7.1.1. Microbiome

The chronic nature of ME/CFS suggests that its continuous pathogenesis involves an altered state of body homeostasis. Numerous reports indicate chronic inflammation, characterised by immunological dysfunction, in ME/CFS. Biomarkers of inflammation and leaky gut syndrome [53], possibly as a result of microbiome disturbance and bacterial translocation, have been highlighted [53,54]. Compellingly, most well-studied inflammatory conditions have been linked to microbial imbalance (dysbiosis) of the human microbiome [53]. Impaired mucosal integrity, shown by serum levels of Immunoglobulin A (IgA) and Immunoglobulin M (IgM) against enterobacteria, may explain both inflammation and the hypersensitivity to food in ME/CFS [55]. Recent gut microbiome studies have found lowered microbial diversity in ME/CFS fecal samples [56,57]. Next-generation sequencing of peripheral blood samples identified a ‘multifactorial microbial component’ that correlated with a disease-severity quality of life measure in ME/CFS patients [58]. Sequencing small ribosomal subunit 16S rRNAs from the collective bacterial population indicated an increase in pro-inflammatory microbiota species and a decrease in anti-inflammatory species [56]. Intracellular pathogens like bacteriophages can drive microbiome dysbiosis by directly interfering with the bacterial molecular biology. Interestingly, a study of monozygotic twins discordant for ME/CFS found an increase in bacteriophages belonging to the tailed double stranded Ribonucleic Acid (dsRNA) *Caudovirales* order [56]. 

As yet, however, there is no general agreement on changes in specific microbiota phyla/genera among reported ME/CFS studies [53,54,55,56,57,58,59,60]. 

#### 7.1.2. Metabolome

ME/CFS metabolomics studies have identified a consistent set of disease characteristics, including increased oxidative stress, depleted amino acids, depleted lipids, TCA cycle and purine metabolites, a reduced folate cycle, and increased sugars [44,61]. Recent studies also show a range of plasma metabolites at abnormally low concentrations in ME/CFS, implying that ME/CFS is a ‘hypometabolic’ syndrome. Naviaux et al. (2016) [44] used broad-spectrum metabolomics to examine metabolites from 63 biochemical pathways and abnormalities were identified in 20 of them. They involved oxidative peroxisomes, mitochondrial metabolism, branch chain amino acids, as well as pathways for sphingolipids, phospholipids and cholesterol. Significantly, despite a diverse heterogeneity of factors leading to the illness of the ME/CFS cohort within the study, the results reflected a lowered cellular metabolic response as a common feature in the patients. It was likened to ‘dauer’, a protective hibernation-like state [44]. 

The suggestion that hypometabolic syndrome is a feature of ME/CFS may be underpinned by changes in expression of key genes in the pathways concerned. Changes in the DNA methylation (epigenetic code) across the promoter region of genes for these pathways could be a key factor. DNA methylation is an epigenetic modification process where DNA methyltransferases add a methyl group to the 5′ position of the cytosine base of 5‘-Cytosine-phosphate-Guanine-3‘ (CpG) dinucleotides [62], and the extent and pattern of this methylation dictates the rate of gene expression [63]. DNA methylation can also recruit transcriptional co-repressors to inhibit the transcription of certain genes [62]. Epigenetic modifications have been suggested to play important roles in inflammatory and autoimmune diseases that share many similarities with ME/CFS [64]. Multiple DNA methylation studies have now shown both hypo-methylation and hyper-methylation at specific gene promoters in ME/CFS patients, including in our own ongoing unpublished studies. We have found that of the changes observed hyper-methylation was proportionally much higher at promoters than throughout the whole genome. Generally, a loss of methylation accounted for most of the genome-wide changes between the ME/CFS study group and the controls. The addition of methyl groups at promoters may contribute to a hypometabolic state by down-regulating the expression of genes involved in key metabolic pathways. The ME/CFS phenotype is linked to differential methylation in genes associated with immune function and cellular metabolism [48,65,66]. For example, a 2017 study [48] detected 12,608 differentially methylated sites predominantly at cellular metabolism genes, changes that also could be related to patient quality of life health scores. Among these, glucocorticoid sensitivity was associated with differential methylation at 13 loci, implicating this process, along with immune and HPA axis dysfunctions, in ME/CFS [48]. A study focusing on CD4+ T-cells of patients affected by ME/CFS, rheumatoid arthritis and multiple sclerosis found differential methylation around the major Histocompatibility complex, class II, DQ beta 1 gene (HLA-DQB1) [67]. HLA-DQB1 encodes a cell surface receptor essential in immune signaling. Overall, these findings align with recent ME/CFS work pointing towards impairment in cellular energy production and immune dysfunction in the patient population. 

#### 7.1.3. Mitochondria

Detailed studies on energy production pathways in ME/CFS implicate dysfunctional mitochondria in the disease pathogenesis. Crucial fatigue symptoms of exercise intolerance and myalgia associated with ME/CFS are shared by patients with primary mitochondrial diseases and known mutations in either nuclear or mitochondrial DNA [68]. A recent study found lower maximal respiration in ME/CFS PBMCs, suggesting a reduced ability to elevate their respiration rate to compensate in times of physiological stress [46], and we have found a similar result in ongoing studies within our patient cohort. Another study showed a reduced abundance of the amino acids that fuel oxidative metabolism via the citric acid cycle in mitochondria. The changed amino acid pattern suggested the functional impairment of a key enzyme, pyruvate dehydrogenase (PDH) [45], supported by the identification of the increased expression of transcripts of kinases that inhibit PDH. Interestingly, myoblasts grown in serum from severe ME/CFS patients showed metabolic adaptations, including increased mitochondrial respiration and lactate secretion [45].

#### 7.1.4. Transient Receptor Potential (TRP) Ion Channels

A recent study has documented extensively the dysfunction of a TRP ion channel and Ca^2+^ mobilisation in ME/CFS [69]. The **T**ransient **R**eceptor **P**otential group **M** (TRPM) subfamily participates in store operated calcium entry (SOCE) in the white matter of the CNS [69]. Previous investigations have proposed that Natural Killer (NK) cells from ME/CFS patients have a significantly reduced expression of TRPM3 and a subsequent reduction in intracellular Ca^2+^ mobilisation [70,71]. Five single nucleotide polymorphisms (SNPs) have now been identified in the TRPM3 gene in ME/CFS patients that may confer susceptibility to the disease [72]. A significant reduction in NK cell cytotoxicity, a Ca^2+^ dependent process, is consistently reported in both severe and moderate ME/CFS [73]. Related to ME/CFS symptomology, TRPM3 ion channels also have a role in the detection of heat and in pain transmission in the CNS [74]. Collectively, these results suggest that disturbed TRPM3 expression or activity may play an important role in the pathophysiology of ME/CFS. 

#### 7.1.5. Genetic Susceptibility

Single Nucleotide Polymorphism (SNP) analyses generally have provided weight to a genetic susceptibility for ME/CFS. Associations but not causative mutations have been found between several SNPs and ME/CFS pathology. An evaluation of 116, 204 SNPs found 65 SNPs associated with ME/CFS that included a glutamate receptor, ionotropic kinase 2 (GRIK2) (decreased expression), and neuronal ‘Per-Arnt-Sim’ (PAS) domain protein 2 (NPAS2) (increased expression) implicating a pathological role for genes involved in glutamatergic neurotransmission and circadian rhythm [75]. Interestingly, transcripts associated with circadian rhythm were identified as significantly changed in our transcriptome study [49]. Most recently, Schlauch et al. evaluated 906, 600 SNPs and found 442 associated with ME/CFS [76]. 

## 8. Has Biomedical Research Informed the Clinic, and Assisted Diagnosis and Treatment?

With a myriad of clinical case definitions to choose from, and a lack of understanding of the fundamental pathophysiology of ME/CFS, coupled with a lack of a definitive molecular biomarker, health practitioners have not had a clear direction for the diagnosis and treatment of ME/CFS. In our experience, this has led to frustration both at the feet of the practitioners and the affected patients, and to a long period of debate among the health profession about the true nature of the illness. It has hindered funding for much needed research and has created inertia for researchers to join the research effort. Excitingly, the last 3–4 years have signaled dramatic change. With growing understanding of the disease biology, and with promising diagnostic tools on the horizon, the outlook is much more promising for both practitioners and patients alike. Current biomedical research is on the cusp of providing tools that can be utilised in the clinic, but already the identification of affected pathways like inflammation and disturbed immunity, or changes in energy production and key metabolic pathways, and the identification of potential biomarkers for those changes are starting to inform the clinic today. Therapeutic interventions have been ‘trial and error’ applications of available drugs but with no universal benefit, and at best very mixed results. We are entering a new phase where the directed design of therapies seems possible. Examples are antioxidants that might target and reduce an energy production insufficiency, and targeted anti-inflammatory drugs to improve impaired central nervous system function. The ultimate goal of the biomedical research is to find the primary cause of the significant downward spiral in health that results in the long-term debilitating illness of ME/CFS.

## 9. Discussion

### Future Directions and Unresolved Questions

While major steps have been made in our understanding of the biological processes involved in ME/CFS, there are important unresolved questions remaining. These include: (i) Is there a genetic susceptibility that causes some individuals, after exposure to a viral or toxic chemical or a traumatic emotional assault, to ‘switch’ into a life-long ME/CFS ‘dauer’ state? (ii) What is the key initial physiological trigger causing the dramatic downward spiral in health? (iii) Are the molecular changes and aberrant energy producing pathways observed a consequence of, rather than the cause, of the disease state? (iv) Is there an unidentified ‘molecular factor’ that facilitates cell alterations, as implied from the effects of ME/CFS serum on healthy cells? (v) Why does the ‘switch’ triggering a physiological response to the initial assault not return to normal, as for most viral illnesses? and, (vi) Why do characteristic frequent relapses occur in the chronic stage of ME/CFS, implying a hypersensitivity to even minor stress? 

ME/CFS was classified originally by the World Health Organization as a neurological disease in 1969 [77], and many symptoms like lack of refreshing sleep and cognitive ‘brain fog’ must be directly related to a poorly functioning brain. We have proposed a neuro-inflammatory model of ME/CFS, in an attempt to describe its unexplained and diverse characteristics and wide array of symptoms [78]. We believe that a centre within the hypothalamus, a cluster of neurons called the paraventricular nucleus that is key in resolving stress, may be critical to the perpetuation of the disease and the relapse/recovery cycles in ME/CFS [78]. One neuroimaging study with ME/CFS patients [78] provides support for this hypothesis, with the discovery of enhanced activated glial cells (a marker for neuro-inflammation) in the limbic system. The degree of glial activation correlated with the severity of ME/CFS symptoms [79]. To test these ideas, targeted advanced imaging studies are needed which will provide a better understanding of the specific mechanisms in the brain that are affected to cause such a severe phenotype in ME/CFS. 

Current global ME/CFS research has focused on identifying changes in physiological and biochemical pathways, with special emphasis on energy metabolism and Ca^2+^ metabolism. More detailed research on the deficiencies coupled with genetic analyses to explain the hypo-metabolism may prove vital in understanding what it is in ME/CFS that sustains the illness and its complex symptoms. With the powerful new analytical tools available to researchers, rapid advancement is being made in our understanding of the underlying biology of ME/CFS, and several promising potential disease biomarkers have been identified. What is now needed are significant follow-up investigation of these markers, with large patient numbers and across different centres. Most importantly, these biomarkers will need to be validated against diseases that share similar features with ME/CFS.

## Figures and Tables

**Figure 1 diagnostics-09-00073-f001:**
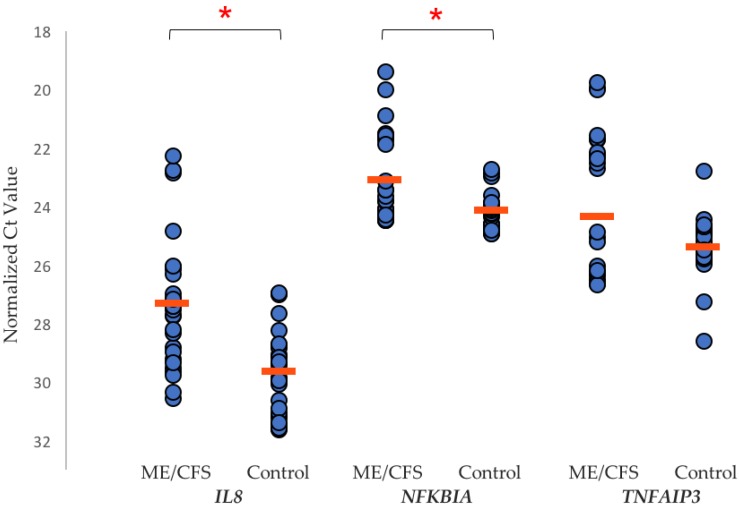
Scatter plot of RT-qPCR assay Ct values (defined below) for *IL8*, *NFKBIA*, and *TNFAIP3*. Each peripheral blood mononuclear cell (PBMC) sample (myalgic encephalomyelitis/chronic fatigue syndrome (ME/CFS) *n* = 10, control *n* = 10) was measured in triplicate, with the mean Ct value for each gene in both ME/CFS and control cohorts shown (orange line). A Ct value is the **R**everse **T**ranscription **q**uantitative **P**olymerase **C**hain **R**eaction (RT-qPCR) amplification cycle at which the gene transcript copy number exceeded the individually calculated baseline threshold level for that gene. Figure taken from our own recently published study [49]. Red bars indicate the mean Ct value in each case. Statistical significance (*p* < 0.05) between the two groups is indicated by the *.

**Figure 2 diagnostics-09-00073-f002:**
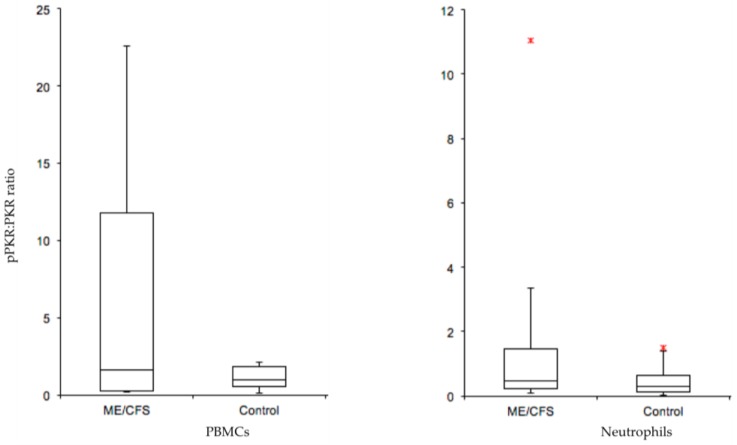
Box-and-whisker plots of ME/CFS and control phosphorylated PKR/protein kinase RNA-activated (pPKR:PKR) ratios in PBMCs and neutrophils. In-house affinity purified antibodies against phosphorylated PKR (active form) and PKR (inactive form) were used to detect the ratio of pPKR:PKR in isolated PBMCs and neutrophils from a matched patient/control ME/CFS study. The median pPKR:PKR ratio is shown, and the interquartile range, maximum and minimum ratio values. Outliers are indicated with a (*). A t-test between ME/CFS and controls gave a P-value of 0.057 in PBMCs (ME/CFS *n* = 9, controls *n* = 9) and 0.142 in Neutrophils (ME/CFS *n* = 9, controls *n* = 10). This figure has been constructed by author ES from data referenced in her PhD thesis, with permission to publish from University of Otago [38].

**Table 1 diagnostics-09-00073-t001:** Biological Pathways affected in ME/CFS.

Affected Biological Pathways	References
Immune/inflammation	[37,38,39,40,41,49]
Cytokine regulation	[37,38,42,43,49]
Metabolic dysregulation	[38,44,45,49]
Mitochondrial dysfunction	[38,45,46,49]
Oxidative stress	[38,39,47,49]
Apoptosis	[38,39,47,49]
Circadian rhythm	[48,49]

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
