# Peer review of "Current Research Provides Insight into the Biological Basis and Diagnostic Potential for Myalgic Encephalomyelitis/Chronic Fatigue Syndrome (ME/CFS)"

_diagnostics, 2019, doi:10.3390/diagnostics9030073_

Reviewer 1 Report

The paper is well written.

1. I think it did put too much emphasis on the term "Fatigue" which is an unfortunate sterotype when the disease was intially named as fatigue is no less an important clinical presentation that the many immunological issues. Notwithstanding, the paper is technicall accurate.
2. It does need some rigerous proofreading. For instance, Multiple Sclerosis is not capatalized and there are a few samll typos, such as ME/CF etc. Once the paper is carefully proofread and correct I think it is good to go!

Author Response

1. We appreciate the point. While fatigue is the core symptom experienced by ME/CFS patients initially it is a symptom shared with many other illnesses and that is why excluding these illnesses is so important to establish a diagnosis of ME/CFS currently. The word fatigue has now been used more sparsely in the manuscript. For example, we have replaced  ‘post exertional fatigue’ with the commonly used more general term ‘post exertional malaise’. We have done exercise physiology studies with ME/CFS alongside multiple sclerosis patients and ME/CFS uniquely perform poorly on re exercise, demonstrating this post exertional malaise.
2. Rigorous proof reading has been done by 3 of the authors. These errors have been noted and corrected in the document.

Reviewer 2 Report

This paper goes through several fields of interest to the understanding of the pathophysiology of CFS/ME. I have some comments:

1. This is not a review in the sense that there is a strategy for collecting published material, and there is no quality criteria or discussions of strengths and limitations

2. The title "New research.." is not justified, since everything that is presented seems that have been published already

3. There are several claims made without documentation - for instance the claim that vaccinations can precipitate the disease - and the statement that CFS/ME is a life-long disease

4. Results from a study of 10 cases is presented, but there are no details on the selection of cases and methods used

Author Response

1. We thought about this question ourselves before submitting. As there were no prescriptions for this on the Journal site we accessed definitions for each type of presentation and it seemed to us it fitted either a narrative review, which does not require a methodological approach nor evaluation criteria for inclusion of publications, or indeed a commentary. However we are happy to re classify it as a ‘Commentary' as we agree on balance that this is the better fit for our manuscript, and have revised the document with the included suggested corrections.  We have used the following definition as a guide from International Journal of Qualitative Studies for Health and Wellbeing "Guidelines for writing a commentary”:

What is a commentary? The goal of publishing commentaries is to advance the research field by providing a forum for varying perspectives on a certain topic under consideration in the journal. The author of a commentary probably has in-depth knowledge of the topic and is eager to present a new and/or unique viewpoint on existing problems, fundamental concepts, or prevalent notions, or wants to discuss the implications of a newly implemented innovation. A commentary may also draw attention to current advances and speculate on future directions of a certain topic, and may include original data as well as state a personal opinion. While a commentary may be critical of an article published in the journal, it is important to maintain a respectful tone that is critical of ideas or conclusions but not of authors.

2. We are happy to change the title to ‘Current research’ as we agree with the above point - the majority of the research presented has been published and “Current research” is a more appropriate descriptor/title. However, some of the results discussed through the document are as-yet unpublished, for example, certain results from the 10 patient/control study, and we have now made this clearer in the document. 
3. References have now been added for the points raised -(i) vaccinations precipitating ME/CFS and (ii) the life-long nature of the illness for a subset of patients.

4. We appreciate the reviewer’s point - we have only stated briefly how our patient cases were selected - using the CCC guidelines. We have now made this statement clearer.

We have also included a statement of the methods used to obtain the results we discuss from this patient/control study. One of the authors is the expert clinician who has vast experience with diagnosing ME/CFS according to clinical guidelines and uses the CCC guidelines that to her give the most accurate diagnosis, along with long patient interviews.